# MarkovScale: Towards Optimal Sequential Scaling at Inference Time

## Abstract

Sequential scaling is a prominent inference-time scaling paradigm, yet its performance improvements are typically modest and not well understood, largely due to the prevalence of heuristic, non-principled approaches that obscure clear optimality bounds. To address this, we propose a principled framework that models sequential scaling as a two-state Markov process. This approach reveals the underlying properties of sequential scaling and yields closed-form solutions for essential aspects, such as the specific conditions under which accuracy is improved and the theoretical upper, neutral, and lower performance bounds. Leveraging this formulation, we develop MarkovScale, a practical system that applies these optimality criteria to achieve a theoretically grounded balance between accuracy and efficiency. Comprehensive experiments across 3 backbone LLMs, 5 benchmarks, and over 20 configurations show that MarkovScale consistently outperforms state-of-the-art parallel and sequential scaling methods, representing a significant step toward optimal and resource-efficient inference in LLMs. The source code will be open upon acceptance at `https://open-upon-acceptance`.

## 1 Introduction

The pursuit of enhanced inference capacity in large language models (LLMs) has long been guided by the scaling law paradigm, which posits that performance improvements are achieved primarily through increased computational resources and enables training on larger datasets with more parameters. However, a recent paradigm shift has emerged toward Inference-time Scaling (Brown et al., 2024; Wu et al., 2024; Snell et al., 2024; Zhang et al., 2025), where performance gains are attained through increasing computation budgets at inference time, without modifying the base model. This approach offers distinct advantages: it preserves the original model architecture, reduces dependency on extensive computational infrastructure, and provides a flexible, efficient alternative to conventional training-based scaling methods. This is particularly valuable for users with limited access to model parameters or high-performance computing resources.

Current research on inference-time scaling methods broadly distinguishes between two approaches: *parallel scaling* and *sequential scaling*. Parallel scaling techniques (e.g., Best-of-N (Cobbe et al., 2021), Majority Voting (Wang et al., 2023; Nguyen et al., 2024)) aggregate multiple independent model outputs to select optimal responses, while sequential methods (e.g., Chain-of-Thought (Wei et al., 2022; Li et al., 2024b), Multi-Round Thinking (Tian et al., 2025)) iteratively refine outputs by incorporating previous reasoning steps into subsequent iterations. Although empirical studies indicate that sequential scaling typically achieves smaller performance gains compared to parallel or hybrid approaches, it offers distinct advantages in token efficiency (Wang et al., 2025a). Moreover, the sequential paradigm proves particularly valuable for interactive inference scenarios like multi-agent debates, where each agent's arguments must build coherently upon previous exchanges. Our research focuses on advancing sequential scaling methods to match or exceed the performance levels achieved through parallel approaches, while preserving their inherent efficiency advantages. We frame this dual objective of maximizing performance while minimizing resource consumption as the fundamental challenge of **scaling optimality**.

The difficulty in achieving optimality with current approaches stems from their reliance on heuristic methods rather than principled statistical foundations. As a result, it is challenging to derive closed-form expressions for performance or token usage (upper and lower) bounds. To address the gap,

we demonstrate that the sequential scaling process can be modeled as a two-state Markov process, where scaling transitions between states correspond to the LLM amending an answer to either a correct or incorrect one. By incorporating the zero-shot probability that the LLM outputs a correct answer at the initial iteration, we can formally express the probability distribution over the two states at any iteration. This allows us to establish closed-form solutions for the bounds on optimality.

Our key contributions are as follows:

- **Principled Formulation**: We develop a novel discrete-time, two-state Markov formulation that provides a probabilistic characterization of sequential scaling dynamics. This framework not only enhances interpretability but also uncovers fundamental properties of sequential scaling. Notably, the performance convergence for a given LLM can be analytically predicted in advance, supporting model evaluation and selection prior to large-scale experimentation.

- **Performance Characterization**: Our analysis reveals that sequential scaling performance is determined by both the model's inherent inference capacity (captured by transition probabilities) and the question-dependent zero-shot accuracy. The formulation enables early-stage calculation of optimal stopping iterations that balance accuracy and efficiency, while also establishing clear criteria for when scaling improves or degrades performance.

- **Practical Implementation**: We develop MarkovScale, a practical system that applies the derived optimality criteria by terminating generation at the theoretical optimal state. Comprehensive evaluation across 3 backbone LLMs, 5 benchmarks, and over 20 configurations demonstrates that MarkovScale outperforms state-of-the-art methods in both parallel and sequential scaling settings.

## 2 RELATED WORK

**Inference-time Scaling for LLM Reasoning.** Inference-time scaling is a promising paradigm for enhancing LLMs in complex reasoning tasks without modifying model parameters (Wei et al., 2022). By scaling more compute at inference time rather than retraining, it offers a lightweight and flexible alternative for improving reasoning performance. Inference-time scaling methods can be broadly categorized into *parallel scaling* and *sequential scaling*. Parallel methods like Self-Consistency (Wang et al., 2022) and Best-of-N (BoN) generate multiple independent outputs during a single forward pass, aggregating results through voting or selection. Sequential methods refine outputs step-by-step, using previous reasoning as context for deeper inference. Although sequential scaling often yields smaller performance gains than parallel or hybrid methods, it offers notable advantages in token efficiency (Wang et al., 2025a), motivating our focus on advancing this direction.

**Sequential Scaling.** Sequential scaling enhances LLM reasoning through multi-step iterations, where each step builds upon prior outputs to enable reflection, refinement, and correction (Madaan et al., 2023). However, naive iterative approaches often suffer from diminishing returns or even performance degradation (Madaan et al., 2023). For example, Multi-round Thinking (Tian et al., 2025) has attempted to improve answer quality by revisiting and refining initial outputs, but gains tend to plateau quickly. This highlights a key insight: simply adding steps is insufficient, and effective sequential scaling requires strategic guidance to ensure meaningful progress. A prominent class of sequential methods builds on LLMs' intrinsic abilities for self-correction and self-verification. ReVISE (Lee et al., 2025) enables models to assess and revise their own reasoning paths, while SETS (Chen et al., 2025) combines sampling, verification, and correction into a unified loop. Self-Refine (Madaan et al., 2023; Chen et al., 2023) adopts a recursive feedback loop, using the same model as generator, critic, and revised. Despite their promise, these methods can be unstable due to LLMs' limited ability to reliably detect and correct their own errors. Additionally, growing attention to budget-awareness has led to methods like s1 (Muennighoff et al., 2025), which control inference cost via decoding-time interventions, and others that study reasoning under output length constraints or penalize overly long reasoning chains (Sun et al., 2025; Aggarwal & Welleck, 2025). Concepts such as "overthinking" (Chen et al., 2024) and "underthinking" (Wang et al., 2025b) further underscore the need for effective and efficient resource allocation in sequential scaling. As a representative approach, Atom of Thoughts (Teng et al., 2025) has decomposed reasoning into an iterative decomposition-contraction process, dedicating computational resources to reasoning directly related to the current atomic question state.

In contrast to prior heuristic or empirically driven methods, our proposed MarkovScale framework provides a principled and theoretically grounded approach to sequential scaling. By modeling the iterative process as a Markov chain, we derive explicit guarantees for optimal stopping and convergence to an inherent accuracy upper bound, addressing the challenges of performance instability and inefficient resource usage observed in earlier work.

## 3 MARKOV SEQUENTIAL SCALING

In this section, we model sequential scaling as a Markov process. This formulation not only reveals the core properties of sequential scaling, but also enables us to derive closed-form conditions under which the scaling proves beneficial.

### 3.1 SEQUENTIAL SCALING AS A MARKOV PROCESS

Sequential scaling is an iterative inference process. Given a question $q$, the LLM produces an answer $o_i$ at the $i^{th}$ iteration using the previous answer $o_{i-1}$ at the $(i-1)^{th}$ iteration as $o_i = f(q, o_{i-1})$. We model the state of each iteration by indicating whether the answer is correct as $X_i \in \{C, W\}$, where $C$ means the answer $o_i$ is correct, $W$ otherwise. This leads to a *two-state Markov Chain*, since

$$\mathbb{P}(X_i|X_{i-1}, X_{i-2}, \cdots, X_0) = \mathbb{P}(X_i|X_{i-1}). \tag{1}$$

Let us define the transition probabilities as follows: 1) $a = P(X_i = W \mid X_{i-1} = C)$: The probability of transitioning from a correct answer to an incorrect one. 2) $b = P(X_i = C \mid X_{i-1} = W)$: The probability of transitioning from incorrect to correct. This gives the transition matrix

$$\mathbf{P} = \begin{bmatrix} 1-a & a \\ b & 1-b \end{bmatrix}.$$

Let $p_0(q) = \mathbb{P}(X_0 = C)$ represent the zero-shot probability of the LLM outputting a correct answer in the initial iteration. The state probability distribution vector $\boldsymbol{\pi}_i$ at the $i^{th}$ iteration is given by:

$$\boldsymbol{\pi}_i = \begin{bmatrix} \mathbb{P}(X_i = C), \mathbb{P}(X_i = W) \end{bmatrix} = \boldsymbol{\pi}_0 \underbrace{\mathbf{PP} \cdots \mathbf{P}}_{i \text{ times}} = \boldsymbol{\pi}_0 \mathbf{P}^i = [p_0, 1-p_0]\mathbf{P}^i \tag{2}$$

### 3.2 PROBABILITY OF CORRECT STATE CONVERGENCE

Let $p_i(q) = \mathbb{P}(X_i = C)$ denote the probability of obtaining a correct answer at the $i^{th}$ iteration. From Eq. (2), we have $p_i = \boldsymbol{\pi}_0 \mathbf{P}^i \mathbf{e}_1$ where $\mathbf{e}_1 = [1, 0]^\top$ is the basis vector for the correct state. We further decompose the $\mathbf{P}$ using Matrix diagonalization as $\mathbf{P} = \mathbf{SDS}^{-1}$, where

$$\mathbf{D} = \begin{bmatrix} 1 & 0 \\ 0 & \lambda \end{bmatrix}, \quad \mathbf{S} = \begin{bmatrix} 1 & a \\ 1 & -b \end{bmatrix}, \quad \mathbf{S}^{-1} = \frac{1}{a+b}\begin{bmatrix} b & a \\ 1 & -1 \end{bmatrix}$$

and $\lambda = 1 - a - b$. This gives

$$\mathbf{P}^i = \frac{1}{a+b}\begin{bmatrix} b + a\lambda^i & a(1-\lambda^i) \\ b(1-\lambda^i) & a + b\lambda^i \end{bmatrix}. \tag{3}$$

The $p_i$ is then calculated as

$$p_i = p_0 \cdot \mathbf{P}^i_{11} + (1-p_0) \cdot \mathbf{P}^i_{21} = \frac{b}{a+b} + \frac{\lambda^i}{a+b}\left(ap_0 - b(1-p_0)\right). \tag{4}$$

### 3.3 BENEFICIAL SCALING CONDITIONS

Previous studies have shown empirically that sequential scaling can sometimes improve performance and, in other cases, lead to degradation (Wang et al., 2025a; Tian et al., 2025). However, these findings remain observational and lack a theoretical explanation, especially regarding the underlying mechanisms and the precise conditions under which scaling is beneficial. As indicated by Eq. (4), the Markov sequential scaling process is determined by the capability of backbone models and the

difficulty of questions. These two major factors are jointly reflected and represented by the zero-shot probability $p_0$ and the transition probabilities $a$ and $b$. This formulation allows us to establish closed-form criteria that specify when sequential scaling will be advantageous, based on these parameters.

To quantify the advantages of sequential scaling, we introduce a benefit function

$$g_i(q) = p_i(q) + \sigma - p_0(q). \tag{5}$$

This measures the improvement in answer quality by comparing iterative performance (i.e., $p_i(q)$) against the baseline (i.e., $p_0(q)$) for a given question $q$ regarding a theoretical bias term $\sigma$. When estimating $p_i(q)$, $\sigma$ is used to account for the complexities of sequential reasoning process, fluctuations, or randomness that can't be entirely modeled by the simplified 2-state Markov chain modeling, together with prediction error margins. $g_i(q)$ yields a value greater than zero when scaling improves answer quality, and a value less than zero upon performance degradation.

Substituting the closed-form expression for $p_i(q)$ from Eq. (4) into $g(q)$, we have

$$g_i(q) = \left[ L + \lambda^i(p_0 - L) \right] - p_0 + \sigma = (L - p_0) + \lambda^i(p_0 - L) + \sigma = (L - p_0)(1 - \lambda^i) + \sigma \tag{6}$$

where $L = \frac{b}{a+b}$ and $\lambda = 1 - a - b$. Since $\lambda \in (0, 1)$, $\lambda^i \to 0$ when $i \to \infty$, the asymptotic benefit

$$\lim_{i \to \infty} g_i(q) = (L - p_0)(1 - 0) + \sigma = L - p_0 + \sigma. \tag{7}$$

The sign of $\lim g(q)$ defines the following three scaling regimes: 1) **Beneficial Scaling** ($\lim g(q) > 0$): Occurs when $p < L + \sigma$ and the scaling monotonically improves confidence; 2) **Detrimental Scaling** ($\lim g(q) < 0$): Occurs when $p > L + \sigma$ and the scaling causes asymptotic degradation; and 3) **Neutral Scaling** ($\lim g(q) = 0$): Occurs when $p = L + \sigma$ and the confidence remains no change. These closed-form conditions enable the selective application of scaling as a criterion as follows:

**Theorem 3.1** (Beneficial Sequential Scaling Criterion). *For a given question q, sequential scaling is **provably beneficial** if and only if:*

$$p_0(q) < \frac{b}{a + b} + \sigma. \tag{8}$$

### 3.4 MODEL SELECTION AND PERFORMANCE BOUND QUANTIFICATION

In addition to identifying when scaling is advantageous, practitioners commonly face two main challenges: 1) selecting the optimal backbone LLMs from available options prior to experimentation, and 2) quantifying how closely a chosen scaling strategy approaches its theoretical upper and lower performance bounds. Our framework offers principled solutions to both of these challenges.

**Model Selection.** For a given candidate model, our closed-form solution (Eq. (4)) allows practitioners to directly estimate its expected convergence accuracy $p_i$ (under naive sequential scaling) even before conducting any experiments, as follows:

**Theorem 3.2** (Expected Convergence Accuracy). *For any question satisfying $p < L$ with $|\lambda| < 1$ where $\lambda = 1 - a - b$, the correctness probability converges exponentially to:*

$$\lim_{i \to \infty} p_i = L = \frac{b}{a + b} \tag{9}$$

*Proof.* From Eq. (4), $p_i = L + \lambda^i(p_0 - L)$, the transient term $\lambda^i(p_0 - L)$ vanishes as $i \to \infty$ since $|\lambda| = |1 - a - b| < 1$. The remaining term $L$ constitutes the fixed point. Yet in practice, The ultimate convergence accuracy will need to account for bias $\sigma$, making convergence at $L + \sigma$. $\square$

This analysis exposes the model's inherent ability for sequential scaling, with $L$ representing the expected accuracy under unbounded scaling. It can serve as a metric for model selection.

**Performance Bounds.** The theorem can also be extended to estimate the performance bounds for a given model. Specifically, after applying naive sequential scaling to a dataset, we obtain:

- **Neutral Bound**: $\bar{L} = \frac{\bar{b}}{\bar{a}+\bar{b}}$, where $\bar{a}$ and $\bar{b}$ are the observed transition probabilities. This represents the baseline convergence point for most scaling strategies and serves as a baseline for selecting strategies (i.e., if $\bar{L}$ is inadequate, alternative strategies should be considered).

- **Upper Bound**: $L^+ = \frac{b^+}{a^+ + b^+}$, where $a^+$ and $b^+$ are the transition probabilities corresponding to Beneficial Scaling questions (as identified by Eq. (7)). This is the theoretical maximum accuracy attainable and can be used to assess how closely a strategy approaches optimal performance.

- **Lower Bound**: $L^- = \frac{b^-}{a^- + b^-}$, where $a^-$ and $b^-$ are the transition probabilities corresponding to Detrimental Scaling questions (as identified by Eq. (7)). This is the worst-case accuracy, representing the performance when a scaling is detrimental, thereby limiting the model's effectiveness.

# 4 OPTIMAL MARKOV SEQUENTIAL SCALING

## 4.1 MARKOVSCALE WITH GATING STRATEGY

Our Markov formulation enables the derivation of principled scaling strategies in closed form. With Theorem 3.1, we can implement a basic gating strategy that completely bypasses scaling when the initial accuracy $p_0(q)$ meets or exceeds the threshold $\frac{b}{a+b} + \sigma$. This automatically filters out questions that fall into the Detrimental or Neutral scaling regimes. We denote this strategy as **MarkovScale**$^t$.

## 4.2 MARKOVSCALE WITH SCALING OPTIMALITY

While MarkovScale$^t$ enables selective scaling to reduce token usage, it is not necessarily optimal for questions within the Beneficial scaling regime. To address this, we define **Scaling Optimality** as an operational state where sequential scaling simultaneously satisfies two conditions: 1) the resulting accuracy surpasses a specified confidence level $\tau \in (0, 1)$, and 2) token usage is minimized. We model this as an optimization problem to determine the optimal iteration count $i^*$ as

$$i^* = \underset{i \in [0, \infty)}{\arg \min} \, p_i \geq \tau, \; \tau \in (0, 1), \, i \in \mathbb{Z}^+. \tag{10}$$

Substituting Eq. (4) into Eq. (10), we have the optimal stopping criterion:

$$\frac{b}{a+b} + \frac{\lambda^i}{a+b} \left( ap_0 - b(1 - p_0) \right) \geq \tau \implies \lambda^i \left( (a+b)p_0 - b \right) \geq (a+b)\tau - b. \tag{11}$$

For the non-trivial case where $ap_0 \neq b(1 - p_0)$ and $\lambda \neq 0$, we obtain the closed-form solution:

$$i^* = \left\lceil \log \left( \frac{|(a+b)\tau - b|}{|ap_0 - b(1 - p_0)|} \right) \log^{-1}(\lambda) \right\rceil. \tag{12}$$

For other special cases, we have (see Appendix B for the derivation process): 1) $i^* = 1$ (Immediate satisfaction), when $\tau \leq \frac{b}{a+b}$; 2) $i^* = \frac{b}{a+b}$ (Uniform convergence), when $ap_0 = b(1 - p_0)$; and 3) $i^* = 1$ (Exact convergence in 1 iteration), when $\lambda = 0$. Scaling optimality can be achieved by comparing $i$ to those values and stopping scaling at the optimal iteration to save token consumption.

### 4.2.1 MARKOVSCALE$^*$: MAXIMUM A POSTERIORI ESTIMATION

To enable optimal scaling, we need to estimate the transition probabilities $a$ and $b$, as well as the initial probability of a correct answer $p_0$. In our approach, we consider the transition probabilities to be model-specific characteristics that reflect a model's ability to revise its answers (i.e., its self-reflection capabilities). These probabilities can be readily estimated as empirical means from historical inference data, such as state transitions observed during training. On the other hand, the initial probability $p_0$ represents the model's zero-shot performance and is dependent on the specific question $q$ being asked. We employ the Maximum a Posteriori (MAP) for $p_0$, which estimates an initial value through offline learning and refines it dynamically through online optimization. We denote this strategy as **MarkovScale**$^*$. The overall workflow is summarized in Algorithm 1.

**Offline Learning of Priors.** We employ an MLP (denoted as $\mathcal{M}_0$) on top of the frozen model to learn a mapping from a given question $q$ to the probability $p_0(q)$ that the LLM will produce a correct answer for $q$. Our training data consists of question–probability pairs $(q, \bar{p})$, where $\bar{p}$ is obtained by aggregating the results of a specific number of generated answers for each question $q$. Once the MLP layer is trained, the offline estimation of $p_0(q)$ can be obtained by

$$\hat{p}_0(q) = \mathcal{M}_0(\text{LLM}(q)). \tag{13}$$

Similarly, we employ additional MLP layers to learn $a$ and $b$, mapping from a question $q$ to the transition probabilities. To construct question–transition probability pairs, we first perform inference on the training dataset using naive multi-round sequential scaling. We then estimate by counting state transitions between inference steps, specifically cases where the LLM shifts from a previously correct to an incorrect answer (normalized by the number of correct states to compute $a$), and from previously incorrect to correct (normalized by the number of incorrect states to compute $b$). After the training data is constructed, we train two MLPs as the estimators of predicting $\hat{a}(q)$ and $\hat{b}(q)$, given by:

---

**Algorithm 1** MarkovScale* with MAP

---

1: Specify the confidence level $\tau$ and maximum iterations $N$, initialize $\alpha_0, \beta_0$ via Eq. (15)
2: **for** $i = 1$ to $N$ **do**
3:     Generate completion $x_i$, observe $s_i$
4:     Update $\alpha_i, \beta_i$ via Eq. (16)
5:     Compute $p_i^{\text{MAP}}$ via Eq. (17)
6:     Evaluate stopping condition via Eq. (11)
7:     **if** $p_i \geq \tau$ **then break**
8:     **end if**
9: **end for**

---

$$\hat{a}(q) = \mathcal{M}_a(\text{LLM}(q)), \;\; \hat{b}(q) = \mathcal{M}_b(\text{LLM}(q)), \tag{14}$$

where $\mathcal{M}_a$ and $\mathcal{M}_b$ are distinct MLPs. Although this formulation is viable, we find that per-question predictions of $\hat{a}$ and $\hat{b}$ exhibit high variance and lack stability. To mitigate this, we aggregate the per-question estimates by averaging across all questions within each dataset, yielding uniform dataset-level transition probabilities $a$ and $b$ used at inference time.

**Online Refinement of Posteriors.** To refine the offline estimation during inference, we need to define the probability density function (PDF) for $p$. Since the LLM's output (either correct or incorrect) for each question can be viewed as a Bernoulli trial, we model $p$ as the Beta distribution over multiple trials. This gives $p \sim Beta(\alpha_0, \beta_0)$, where the parameters are initialized as

$$\alpha_0 = \hat{p}_0 \cdot \gamma, \quad \beta_0 = (1 - \hat{p}_0) \cdot \gamma. \tag{15}$$

Here, $\gamma > 0$ controls prior strength. For the $i^{th}$ iteration, we update the posterior as:

$$\alpha_i = \alpha_{i-1} + \phi(o_i), \quad \beta_i = \beta_{i-1} + (1 - \phi(o_i)) \tag{16}$$

where the $\phi(o_i)$ is a function that estimates the quality of the $i^{th}$ answer. This function can be straightforwardly implemented using commonly available verifiers, such as process reward models (PRMs). The MAP estimation is then computed as:

$$p_i^{\text{MAP}} = \begin{cases} \frac{\alpha_i}{\alpha_i + \beta_i}, & \text{if } \alpha_i \leq 1 \text{ or } \beta_i \leq 1, \\ \frac{\alpha_i - 1}{\alpha_i + \beta_i - 2}, & \text{otherwise.} \end{cases} \tag{17}$$

### 4.2.2 MARKOVSCALE$^0$: TRAINING-FREE ESTIMATION

While the MAP estimation method presented in Section 4.2.1 is effective, it relies on offline training of the initial probability predictor $\hat{p}_0$ and transition probability predictors $\hat{a}$ and $\hat{b}$. This introduces additional training overhead and model-specific dependencies. To mitigate these dependencies, we propose **MarkovScale$^0$**, a fully training-free variant. Concretely, we initialize the Beta distribution parameters using heuristic priors for $p_0$. For transition probabilities $a$ and $b$, we apply a minimum number of inference rounds for all questions within each dataset, using these samples to calculate the transition probabilities on existing samples statistically. This serves as an approximated empirical estimation of $\hat{a}$ and $\hat{b}$. We then apply the same online refinement strategy described in Eq.(16) and Eq.(17), where $\phi(o_i)$ is computed using PRMs. This approach leverages the iterative nature of sequential scaling to bootstrap probability estimation from scratch, without any model training. As a result, MarkovScale$^0$ is fully adaptive across diverse models.

Table 1: Performance comparisons of different inference-time scaling methods, where the results are reported in accuracy (%). Each benchmark column shows results for $N \in \{8, 16, 32, 64\}$.

| Benchmark | MATH-500 | | | | GSM8K | | | | AIME 2024 | | | | AIME 2025 | | | | AMC 2023 | | | |
|---|---|---|---|---|---|---|---|---|---|---|---|---|---|---|---|---|---|---|---|---|
| $N$ | 8 | 16 | 32 | 64 | 8 | 16 | 32 | 64 | 8 | 16 | 32 | 64 | 8 | 16 | 32 | 64 | 8 | 16 | 32 | 64 |
| **DeepSeek-R1-Distill-Llama-8B** | | | | | | | | | | | | | | | | | | | | |
| Base (No Scaling) | 75.2 | | | | 71.1 | | | | 43.3 | | | | 13.3 | | | | 72.5 | | | |
| Budget Forcing | 75.8 | 75.8 | 75.8 | 75.8 | 86.9 | 86.9 | 86.9 | 86.9 | 23.3 | 23.3 | 23.3 | 23.3 | 23.3 | 23.3 | 23.3 | 23.3 | 65.0 | 65.0 | 65.0 | 65.0 |
| Atom of Thoughts | 33.0 | 33.0 | 33.0 | 33.0 | 50.9 | 50.9 | 50.9 | 50.9 | 16.7 | 16.7 | 16.7 | 16.7 | 6.7 | 6.7 | 6.7 | 6.7 | 35.0 | 35.0 | 35.0 | 35.0 |
| Best-of-N | 82.0 | 83.0 | 83.8 | 84.6 | 74.4 | 75.7 | 76.3 | 76.9 | 43.3 | 50.0 | 53.3 | 66.7 | 26.7 | 30.0 | 30.0 | 33.3 | 75.0 | 75.0 | 80.0 | 82.5 |
| Self-Consistency | 88.4 | 90.2 | 91.2 | 91.8 | 81.8 | 84.6 | 86.7 | 87.5 | 53.3 | 50.0 | 50.0 | 56.7 | 23.3 | 23.3 | 23.3 | 23.3 | 77.5 | 80.0 | 75.0 | 80.0 |
| MR-Thinking | 85.4 | 88.2 | 86.4 | 88.2 | **90.5** | 91.2 | 90.9 | 90.8 | 43.3 | 50.0 | 40.0 | 56.7 | 33.3 | 40.0 | 26.7 | 30.0 | **80.0** | 82.5 | 85.0 | 82.5 |
| ESC | 88.4 | 90.2 | 91.0 | 91.4 | 81.8 | 84.6 | 86.7 | 87.2 | 53.3 | 50.0 | 50.0 | 56.7 | 23.3 | 23.3 | 23.3 | 23.3 | 77.5 | 80.0 | 75.0 | 80.0 |
| ASC | 88.4 | 90.0 | 91.0 | 91.4 | 81.8 | 84.5 | 86.2 | 86.9 | 53.3 | 50.0 | 50.0 | 56.7 | 23.3 | 23.3 | 23.3 | 23.3 | 77.5 | 80.0 | 75.0 | 80.0 |
| DSC | 84.0 | 89.4 | 86.8 | 91.0 | 81.8 | 84.5 | 86.2 | 86.9 | 53.3 | 50.0 | 50.0 | 56.7 | 13.3 | 13.3 | 23.3 | 23.3 | 77.5 | 80.0 | 75.0 | 80.0 |
| MarkovScale$^t$ (Ours) | **90.2** | **92.0** | **92.4** | **93.2** | 89.7 | **92.0** | **93.2** | **93.9** | **60.0** | **73.3** | **73.3** | **76.7** | **40.0** | **43.3** | **43.3** | **43.3** | 80.0 | **87.5** | **95.0** | 92.5 |
| MarkovScale$^*$ (Ours) | **90.2** | **92.0** | **92.4** | **93.2** | 89.7 | **92.0** | **93.2** | **93.9** | **60.0** | **73.3** | **73.3** | **76.7** | **40.0** | **43.3** | **43.3** | **43.3** | 80.0 | **87.5** | **95.0** | 92.5 |
| MarkovScale$^0$ (Ours) | **90.2** | **92.0** | **92.4** | **93.2** | 89.7 | **92.0** | **93.2** | **93.9** | **60.0** | **73.3** | **73.3** | **76.7** | **40.0** | **43.3** | **43.3** | **43.3** | 80.0 | **87.5** | **95.0** | 92.5 |
| **DeepSeek-R1-Distill-Qwen-7B** | | | | | | | | | | | | | | | | | | | | |
| Base (No Scaling) | 89.6 | | | | 87.0 | | | | 53.3 | | | | 36.7 | | | | 82.5 | | | |
| Budget Forcing | 85.8 | 85.8 | 85.8 | 85.8 | 90.7 | 90.7 | 90.7 | 90.7 | 36.7 | 36.7 | 36.7 | 36.7 | 30.0 | 30.0 | 30.0 | 30.0 | 80.0 | 80.0 | 80.0 | 80.0 |
| Atom of Thoughts | 46.8 | 46.8 | 46.8 | 46.8 | 77.7 | 77.7 | 77.7 | 77.7 | 36.7 | 36.7 | 36.7 | 36.7 | 33.3 | 33.3 | 33.3 | 33.3 | 70.0 | 70.0 | 70.0 | 70.0 |
| Best-of-N | **94.8** | 93.8 | 94.4 | 94.0 | 92.4 | 93.5 | 93.1 | 94.0 | 70.0 | **76.7** | 70.0 | **80.0** | 43.3 | 46.7 | 50.0 | **53.3** | **95.0** | **95.0** | **95.0** | **95.0** |
| Self-Consistency | 93.4 | 93.4 | 93.2 | 93.4 | 90.1 | 90.8 | 90.9 | 91.2 | 60.0 | 66.7 | 63.3 | 60.0 | 40.0 | 36.7 | 40.0 | 40.0 | 85.0 | 87.5 | 90.0 | 92.5 |
| MR-Thinking | 91.2 | 91.6 | 91.4 | 92.0 | 88.4 | 88.3 | 88.4 | 88.2 | 56.7 | 53.3 | 63.3 | 70.0 | 40.0 | 43.3 | 33.3 | 40.0 | 87.5 | 87.5 | 90.0 | 90.0 |
| ESC | 93.4 | 93.4 | 93.4 | 93.4 | 90.1 | 90.8 | 90.9 | 91.1 | 60.0 | 66.7 | 63.3 | 60.0 | 40.0 | 36.7 | 40.0 | 40.0 | 85.0 | 87.5 | 90.0 | 92.5 |
| ASC | 93.4 | 93.4 | 93.2 | 93.4 | 90.1 | 90.8 | 90.9 | 91.1 | 60.0 | 66.7 | 63.3 | 60.0 | 40.0 | 36.7 | 40.0 | 40.0 | 85.0 | 87.5 | 90.0 | 92.5 |
| DSC | 93.2 | 93.2 | 93.0 | 93.2 | 90.1 | 90.7 | 90.9 | 91.1 | 60.0 | 66.7 | 63.3 | 60.0 | 40.0 | 36.7 | 40.0 | 40.0 | 82.5 | 87.5 | 87.5 | 90.0 |
| MarkovScale$^t$ (Ours) | 94.6 | **95.0** | **95.0** | **95.6** | **93.3** | **94.1** | **94.3** | **94.6** | 70.0 | 70.0 | **73.3** | 73.3 | **53.3** | **56.7** | 50.0 | 46.7 | 92.5 | **95.0** | **95.0** | **95.0** |
| MarkovScale$^*$ (Ours) | 94.6 | **95.0** | **95.0** | **95.6** | **93.3** | **94.1** | **94.3** | **94.6** | 70.0 | 70.0 | **73.3** | 73.3 | **53.3** | **56.7** | 50.0 | 46.7 | 92.5 | **95.0** | **95.0** | **95.0** |
| MarkovScale$^0$ (Ours) | 94.6 | **95.0** | **95.0** | **95.6** | **93.3** | **94.1** | **94.3** | **94.6** | 70.0 | 70.0 | **73.3** | 73.3 | **53.3** | **56.7** | **53.3** | 46.7 | 92.5 | **95.0** | **95.0** | **95.0** |
| **QwQ-32B** | | | | | | | | | | | | | | | | | | | | |
| Base (No Scaling) | 94.2 | | | | 94.8 | | | | 70.0 | | | | 56.7 | | | | 85.0 | | | |
| Budget Forcing | 92.2 | 92.2 | 92.2 | 92.2 | 87.6 | 87.6 | 87.6 | 87.6 | **73.3** | 73.3 | 73.3 | 73.3 | **70.0** | 70.0 | 70.0 | 70.0 | 67.5 | 67.5 | 67.5 | 67.5 |
| Atom of Thoughts | 56.0 | 56.0 | 56.0 | 56.0 | 89.0 | 89.0 | 89.0 | 89.0 | 43.3 | 43.3 | 43.3 | 43.3 | 20.0 | 20.0 | 20.0 | 20.0 | 80.0 | 80.0 | 80.0 | 80.0 |
| Best-of-N | 94.6 | 94.6 | 95.2 | 94.8 | **95.8** | 95.5 | 95.5 | 95.8 | **73.3** | 73.3 | 73.3 | 76.7 | 66.7 | 63.3 | 63.3 | 66.7 | **95.0** | **97.5** | **97.5** | **97.5** |
| Self-Consistency | 95.2 | 95.2 | 95.0 | 95.4 | 95.6 | 95.5 | 95.7 | 95.8 | 70.0 | 66.7 | 70.0 | 70.0 | 66.7 | 60.0 | 63.3 | 63.3 | 92.5 | 92.5 | 92.5 | 92.5 |
| MR-Thinking | 94.6 | 94.4 | 94.4 | 93.8 | 95.3 | **95.6** | 95.4 | 94.8 | 70.0 | 70.0 | 63.3 | 73.3 | 53.3 | 56.7 | 53.3 | 56.7 | 87.5 | 80.0 | 75.0 | 80.0 |
| ESC | 95.2 | 95.2 | 95.0 | 95.4 | 95.6 | 95.5 | 95.7 | 95.8 | 70.0 | 66.7 | 70.0 | 70.0 | 66.7 | 60.0 | 60.0 | 60.0 | 92.5 | 92.5 | 92.5 | 92.5 |
| ASC | 95.2 | 95.2 | 95.2 | 95.4 | 95.6 | 95.5 | **95.8** | **95.9** | 70.0 | 66.7 | 70.0 | 70.0 | 66.7 | 60.0 | 63.3 | 63.3 | 92.5 | 92.5 | 92.5 | 92.5 |
| DSC | 95.0 | 94.4 | 94.6 | 94.6 | 95.2 | 95.1 | 95.1 | 95.3 | 70.0 | 66.7 | 70.0 | 70.0 | 66.7 | 60.0 | 63.3 | 63.3 | 92.5 | 92.5 | 92.5 | 92.5 |
| MarkovScale$^t$ (Ours) | **96.6** | 96.6 | 96.4 | **96.6** | 95.5 | **95.6** | 95.6 | 95.6 | **73.3** | **76.7** | **80.0** | **83.3** | **70.0** | **73.3** | **76.7** | **76.7** | 87.5 | 87.5 | 87.5 | 87.5 |
| MarkovScale$^*$ (Ours) | **96.6** | 96.6 | **96.6** | **96.6** | 95.4 | **95.6** | 95.6 | 95.6 | **73.3** | **76.7** | **80.0** | **83.3** | **70.0** | **73.3** | **76.7** | **76.7** | 87.5 | 87.5 | 87.5 | 87.5 |
| MarkovScale$^0$ (Ours) | **96.6** | **96.8** | **96.6** | **96.6** | 95.5 | **95.6** | 95.6 | 95.6 | **73.3** | **76.7** | **80.0** | **83.3** | **70.0** | **73.3** | **76.7** | **76.7** | 87.5 | 87.5 | 87.5 | 90.0 |

## 5 EXPERIMENTS

### 5.1 EXPERIMENTAL SETUP

**Benchmarks.** We adopt 5 challenging reasoning benchmarks: (1) **MATH-500** (Hendrycks et al., 2021), with 500 high school competition problems; (2) **GSM8K** (Cobbe et al., 2021), with 8.5K grade school problems; (3) **AIME 2024** (MAA, 2024) and (4) **AIME 2025**[1], each containing 30 pre-Olympiad level math problems; (5) **AMC 2023**[2], with intermediate-level problems from American math competitions. These diverse testbeds assess scaling performance across difficulty levels.

**Backbone Models.** We utilize widely adopted open-source LLMs as backbone models. Specifically, we use **DeepSeek-R1-Distill-Qwen-7B** (Guo et al., 2025) and **DeepSeek-R1-Distill-Llama-8B** (Guo et al., 2025) as distilled reasoning models with different architectures. To assess performance in high-capacity, large-scale settings, we also include **QwQ-32B** (Yang et al., 2025).

**Baseline Methods.** We evaluate our MarkovScale against the following competitive inference-time scaling methods: (1) **Budget-Forcing** (Muennighoff et al., 2025), a simple budget-constrained sequential scaling strategy; (2) **MR-Thinking** (Tian et al., 2025), which iteratively rethinks its pre-

---

[1] https://huggingface.co/datasets/math-ai/aime25.
[2] https://huggingface.co/datasets/math-ai/amc23.

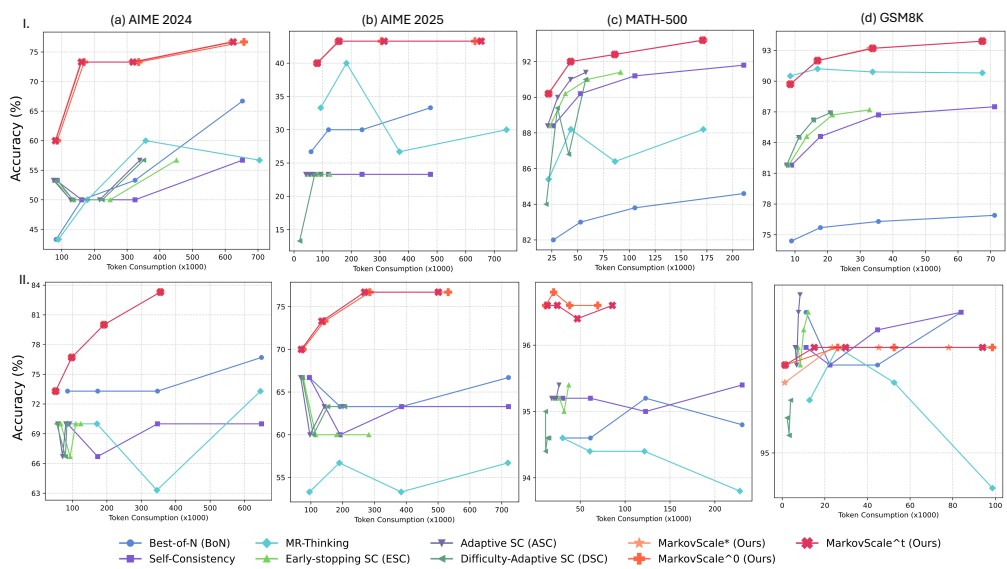

Figure 1: Scaling performance (accuracy vs. average token consumption) of different methods, using (i) DeepSeek-R1-Distill-Llama-8B (top row) and (ii) QwQ-32B (bottom row) as backbones.

vious answer round by round; (3) **Atom of Thoughts (AoT)**[3] (Teng et al., 2025), which adopts an iterative decomposition-contraction process for scaling; (4) **Best-of-N (BoN)** (Cobbe et al., 2021), which generates multiple outputs in parallel and selects the best answer based on external verifier scoring; (5) **Self-Consistency (SC)** (Wang et al., 2023), which relies on majority voting over multiple sampled generations; (6) **Adaptive Self-Consistency (ASC)** (Aggarwal et al., 2023): which stops sampling when existing generations establish a clear majority judged by a lightweight stopping criterion; (7) **Early-stopping Self-Consistency (ESC)** (Li et al., 2024a), which enhances SC with an early stopping strategy; and (8) **Difficulty-Adaptive Self-Consistency (DSC)** (Wang et al., 2024), which leverages difficulty information of questions to allocate budgets adaptively.

**Implementation Details.**   To implement MarkovScale*, we generate 60 answers per MATH-500 training question to estimate zero-shot correctness probabilities. They serve as training data for the MLP predictor, which is trained for 30 epochs with a learning rate of 1e-5. We employ Qwen2.5-Math-PRM-7B as the verifier. In MarkovScale$^0$, we initialize the Beta distribution parameters using empirical priors: $\alpha_0 = 9$ and $\beta_0 = 1$. We define $N$ as the maximum number of sequential iterations or the number of parallel samples allowed during generation, and vary $N$ in $\{8, 16, 32, 64\}$. To ensure a fair comparison, we use a fixed decoding temperature of 0.7 and a top-$p$ value of 0.95 for all methods. All experiments are conducted using 4×H20 GPUs. We report both accuracy and token count to evaluate the effectiveness and efficiency of inference-time scaling, respectively.

### 5.2 PERFORMANCE COMPARISON AND ACCURACY-TOKEN CONSUMPTION TRADEOFF

Table 1 reports quantitative results across various backbone and benchmark configurations, while Figure 1 shows scaling comparisons of different methods in terms of accuracy relative to token consumption. Together, they highlight the effectiveness of MarkovScale in optimizing the tradeoff between performance and efficiency, as evidenced by the following findings:

**Superior Accuracy Across Backbone Models.**   As shown in Table 1, MarkovScale consistently outperforms baselines by a significant margin across all tested backbones. For example, MarkovScale improves accuracy by an average of 19.7% on DeepSeek-R1-Distill-Llama-8B. Even on the stronger QwQ-32B model, the average accuracy gain is 7.7%. These results highlight MarkovScale's generalization ability for inference-time scaling, regardless of the underlying model's archi-

---

[3]Experiments are conducted using the official implementation at `https://github.com/qixucen/atom`. We observed that the current codebase contains an issue in the AoT depth configuration, resulting in the suboptimal performance reported in Table 1.

Table 2: Ablation study results of MarkovScale (Deepseek-R1-Distill-Qwen7B as the backbone with $N = 8$) across different benchmarks. "Acc." denotes accuracy (%), "Toks." indicates token count.

| Baseline Method | MATH-500 | | GSM8K | | AIME 2024 | | AIME 2025 | | AMC 2023 | |
|---|---|---|---|---|---|---|---|---|---|---|
| | Acc. | Toks. ($\downarrow$) | Acc. | Toks. ($\downarrow$) | Acc. | Toks. ($\downarrow$) | Acc. | Toks. ($\downarrow$) | Acc. | Toks. ($\downarrow$) |
| MarkovScale* (full) | 95.0 | **17114** | 93.9 | 4238 | 63.3 | 41192 | 50.0 | **51991** | 92.5 | **16220** |
| *w/o* Scaling Gating (SG) | 95.0 | 17515 | 93.9 | **2801** | 63.3 | **40151** | 50.0 | 52938 | 92.5 | 16235 |
| *w/o* Optimal Stopping (OS) | 95.0 | 20123 | 93.9 | 5056 | 63.3 | 41906 | 50.0 | 54772 | 92.5 | 17962 |
| *w/o* OS + SG | 91.2 | 21396 | 88.4 | 4792 | 56.7 | 78432 | 40.0 | 86568 | 87.5 | 36780 |

tecture. Moreover, our results show that MarkovScale demonstrates robust performance across R1-Distill-Qwen-2.5-7B, R1-Distill-Llama-3.1-8B, and QwQ-32B. This consistency (from 7B to 32B) validates that the proposed method generalizes effectively across different model scales.

**Distinct Benefits of MarkovScale Variants.** Table 1 shows that all three variants of MarkovScale perform well. The simplest of these to implement is MarkovScale$^t$. While MarkovScale$^0$ and MarkovScale$^*$ have comparable accuracy, the training-free MarkovScale$^0$ typically uses more tokens. This is likely because MarkovScale$^0$ requires the LLM to meet a higher internal confidence threshold, often needing more iterations. This highlights a practical tradeoff between the effort of training a lightweight predictor and the efficiency of saving inference tokens.

**Pareto-Optimal Tradeoff Between Accuracy and Cost.** MarkovScale consistently identifies an optimal stopping point in the scaling process, beyond which additional iterations lead to diminishing returns in accuracy. This behavior is especially beneficial for larger models like QwQ-32B, where unnecessary scaling can be costly. As shown in Figure 1, MarkovScale achieves higher accuracy at significantly lower token budgets compared to baselines. For instance, on MATH-500, it attains 97% accuracy while saving about 70% of tokens relative to the MR-Thinking baseline. Similar trends are observed on AIME 2024 and AIME 2025, respectively. These results demonstrate that MarkovScale achieves significant Pareto optimality in the accuracy-cost tradeoff.

**Robust Scaling with Varying Budgets.** MarkovScale shows remarkable robustness to scaling budgets. As reported in Table 1, it consistently maintains state-of-the-art performance across most configurations when varying the scaling budget $N$ within $\{8, 16, 32, 64\}$. Most importantly, it achieves these gains while also reducing token usage by 5% to 70%, depending on the specific configuration. This highlights MarkovScale's potential to efficiently adapt to inference constraints.

### 5.3 ABLATION STUDIES ON SCALING OPTIMALITY CONTRIBUTORS

Table 2 presents an ablation study of the MarkovScale on the DeepSeek-R1-Distill-Qwen-7B with $N = 8$ across different benchmarks. Two main findings emerge: 1) **Effectiveness of Optimal Stopping**: Compared to the variant without the optimal stopping (w/o OS) strategy defined in Eq. (12), the full MarkovScale$^*$ significantly reduces token usage while maintaining the same accuracy. For example, on MATH-500, the full MarkovScale$^*$ uses 17,114 tokens, a 17.6% reduction from the w/o OS variant's 20,123 tokens. Similar trends are also observed on other benchmarks. These results confirm that the proposed optimal stopping strategy prevents unnecessary scaling steps, leading to substantial efficiency gains without sacrificing performance. 2) **Role of Scaling Gating (SG)**: The variant of MarkovScale$^*$ without both optimal stopping and scaling gating (w/o OS+SG) performs much inferior in terms of accuracy and token usage. For instance, on GSM8K, the full MarkovScale$^*$ achieves 93.9% accuracy with 4,238 tokens, while the variant of w/o OS+SG uses 4,792 tokens to obtain 88.4% accuracy, a clear drop in effectiveness and efficiency. These results support Theorem 3.1, validating that the scaling gating is an essential principled strategy for optimal scaling.

### 5.4 VALIDITY AND EFFECTIVENESS OF THE MARKOV FORMULATION AND BEYOND

To assess the validity and effectiveness of the Markov formulation, we compare its theoretically derived bounds with observed empirical results to address the following questions:

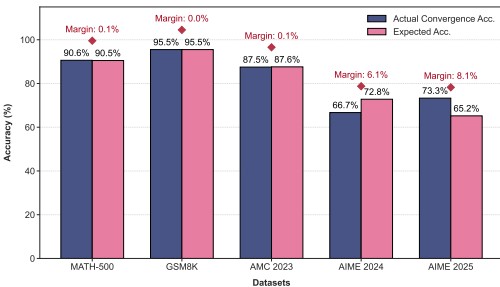

Figure 2: Comparison between expected accuracies and actual convergence results.

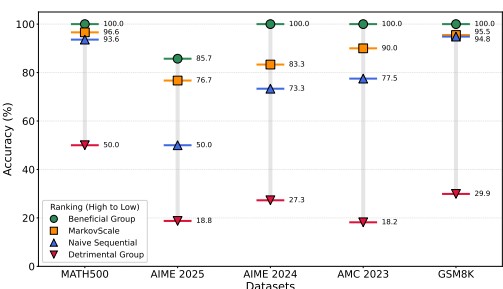

Figure 3: Performance of MarkovScale with respect to the theoretical bounds.

**Can the expected accuracy be predicted?** Theorem 3.2 presents theoretical accuracy limits for naive sequential scaling methods, dependent on the backbone model's capacity as reflected in the transition probabilities. Figure 2 compares these theoretical accuracies with actual convergence results using the Deepseek-R1-Distill-Qwen7B backbone across multiple benchmarks. The theoretical predictions closely match the empirical accuracies, with an average mean absolute error (MAE) of just 2.88%. Remarkably, for three out of five datasets, the prediction error is negligible ($\leq 0.1\%$), highlighting the robustness of our Markov formulation. Since these theoretical accuracies can be computed in advance, they provide useful predictions for expected performance and can inform decisions on whether large-scale experimentation is warranted beforehand.

**Can the theoretical bounds be used to assess performance?** In Section 3.4, we introduce methods for determining theoretical neutral, upper, and lower accuracy bounds. To evaluate MarkovScale's relative performance, we compare its actual accuracy against these bounds. As illustrated in Figure 3, MarkovScale consistently approaches the theoretical upper bound, maintaining a clear advantage over both the neutral and lower bounds. This not only demonstrates the effectiveness of MarkovScale, but also provides a practical framework for assessing future sequential scaling methods by offering clear insight into how close they come to optimal performance, and whether they surpass average (neutral bound) or worst-case (lower bound) scenarios.

## 6 CONCLUSION

This paper introduces MarkovScale, a principled framework for optimal inference-time scaling in large language models (LLMs). By modeling the sequential scaling process as a discrete-time Markov chain, our work establishes the first theoretical foundation and delivers practical algorithms with provable optimality guarantees. MarkovScale addresses a key limitation of existing methods, which rely on heuristic token expansion and suffer from inefficient or unpredictable tradeoffs between accuracy and computational cost. By uncovering the underlying structure of sequential scaling, MarkovScale also opens new avenues for analyzing self-correction dynamics in LLMs, such as the "overthinking" phenomenon, offering valuable insights for complex reasoning tasks.

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

## A  STATEMENT ON THE USE OF LLMS

All content in this paper was drafted by the authors. LLMs like GPT were used solely for error correction and typo checking, as well as occasional rephrasing when necessary. All content is original.

## B  DETAILED SOLUTION DERIVATIONS

Here we provide a detailed solution derivation process of Eq. (11):

$$((a + b)p - b) \cdot \lambda^i \geq (a + b)\tau - b,$$

to find the minimum integer solution $i$ that satisfies the inequality. We follow the previous notations, where $\lambda = 1 - a - b \in (0, 1)$ and $L = \frac{b}{a+b}$.

**Case 1: The left side of the equation equals zero (i.e., $(a + b)p - b = 0$).**

We have

$$(a + b)p - b = 0 \Rightarrow p = \frac{b}{a + b} = L.$$

The inequality becomes:

$$0 \cdot \lambda^i \geq (a + b)\tau - b \Rightarrow 0 \geq (a + b)\tau - b \Rightarrow \frac{b}{a + b} \geq \tau \Rightarrow L \geq \tau.$$

Therefore, we have the solution for Case 1:

- If $p = L \geq \tau$, any $i$ satisfies the inequality (therefore, minimum $i = 1$).
- If $p = L < \tau$, no $i$ satisfies the inequality.

**Case 2: The right side of the equation equals zero (i.e., $(a + b)\tau - b = 0$).**

We have:

$$(a + b)\tau - b = 0 \Rightarrow \tau = \frac{b}{a + b} = L.$$

The inequality becomes:

$$[(a + b)p - b]\lambda^i \geq 0.$$

Therefore, we have the solution for Case 2 as

- When $(a + b)p - b \geq 0$ (i.e., $p > L$):

$$\lambda^i \geq 0.$$

This holds for all $i$ since $\lambda > 0$ (minimum $i = 1$).

- When $(a + b)p - b < 0$ (i.e., $p < L$):

$$\lambda^i \leq 0.$$

Never holds since $\lambda^i > 0$ for all $i$ (no solution).

**Case 3: Signs of LHS and RHS terms align. Now, logarithm is applicable (i.e., $\frac{(a+b)\tau-b}{(a+b)p-b} > 0$).**

**Case 3.1:** $(a + b)p - b > 0$ **and** $(a + b)\tau - b > 0$**.** The inequality becomes:

$$\lambda^i \geq \frac{(a + b)\tau - b}{(a + b)p - b} = \frac{\tau - L}{p - L}.$$

Taking logarithms ($\lambda \in (0, 1) \Rightarrow \lambda^i > 0$ and $\frac{\tau - L}{p - L} > 0$) and moving $\log \lambda$ to the right-hand side ($\log \lambda < 0$):

$$i \leq \frac{\log\left(\frac{\tau - L}{p - L}\right)}{\log \lambda}.$$

There are two subcases:

- When $p > \tau > L$:

$$\frac{\tau - L}{p - L} \in (0, 1) \Rightarrow \log\left(\frac{\tau - L}{p - L}\right) < 0,$$
$$\text{since } \log \lambda < 0 \Rightarrow i \leq \text{positive number.}$$

Thus all $0 < i \leq \frac{\log\left(\frac{\tau - L}{p - L}\right)}{\log \lambda}$ satisfy the inequality (minimum $i = 1$).

- When $\tau > p > L$:

$$\frac{\tau - L}{p - L} > 1 \Rightarrow \log\left(\frac{\tau - L}{p - L}\right) > 0,$$
$$\text{since } \log \lambda < 0 \Rightarrow i \leq \text{negative number.}$$

No solution exists.

**Case 3.2:** $(a + b)p - b < 0$ **and** $(a + b)\tau - b < 0$**.** The inequality direction reverses:

$$\lambda^i \leq \frac{(a + b)\tau - b}{(a + b)p - b} = \frac{\tau - L}{p - L}.$$

There are two subcases:

- When $p < \tau < L$:

$$\frac{\tau - L}{p - L} \in (0, 1).$$

Similarly, taking logarithms, we get:

$$i \geq \frac{\log\left(\frac{\tau - L}{p - L}\right)}{\log \lambda} > 0.$$

The minimum $i$ is the ceiling $\left\lceil \frac{\log\left(\frac{\tau - L}{p - L}\right)}{\log \lambda} \right\rceil$.

- When $\tau < p < L$:

$$\frac{\tau - L}{p - L} > 1.$$

Since $\lambda^i < 1$ for all $i \geq 1$, no solution exists.

**Case 4: Signs of LHS and RHS differ. Logarithm is not applicable (i.e., $\frac{(a+b)\tau - b}{(a+b)p - b} < 0$).**

**Case 4.1:** $(a + b)p - b < 0$ **and** $(a + b)\tau - b > 0$**. The inequality becomes:**

$$\lambda^i \leq \frac{(a + b)\tau - b}{(a + b)p - b} = \frac{\tau - L}{p - L} < 0.$$

No solution exists since $\lambda^i > 0$ for all $i$.

**Case 4.2:** $(a + b)p - b > 0$ **and** $(a + b)\tau - b < 0$**. The inequality becomes:**

$$\lambda^i \geq \frac{(a + b)\tau - b}{(a + b)p - b} = \frac{\tau - L}{p - L} < 0.$$

This holds for all $i \geq 1$ since $\lambda^i > 0 > \frac{\tau - L}{p - L}$ (minimum $i = 1$).

## C   COMPARISON OF DIFFERENT MARKOVSCALE VARIANTS

Table 3 provides a side-by-side comparison of the three MarkovScale variants in terms of accuracy, token usage, and practical deployment considerations. Our findings and insights are as follows: 1) MarkovScale[t], which applies the simplest first-round gating rule, performs surprisingly close to the more sophisticated variants despite its minimal design, making it an appealing choice when implementation simplicity is a priority. 2) MarkovScale[0], which is entirely training-free, delivers consistently strong accuracy and occasionally even surpasses MarkovScale[*], though it typically incurs higher inference-time token costs. 3) In contrast, MarkovScale[*] leverages trained predictors to reduce inference-time computation, and consequently tends to be more token-efficient across most settings. This highlights a practical tradeoff: MarkovScale[0] avoids any training overhead but spends more compute at inference, whereas MarkovScale[*] invests compute upfront during predictor training to achieve more efficient inference-time usage.

## D   HYPERPARAMETER ANALYSES

MarkovScale mainly utilizes two hyperparameters: 1) stopping threshold $\tau$, which dictates the required accuracy probability before the inference process stops scaling, and 2) prior strength factor $\gamma$ (only used in MarkovScale[*]), which controls the weight of the offline prior $\hat{p}_0$ during the online MAP update of the Beta distribution parameters.

Table 4 presents a sensitivity study of MarkovScale[*] as the hyperparameter $\tau$ is varied over $[0.90, 0.99]$. The results show that the optimal choice of $\tau$ is correlated with benchmark difficulty: harder tasks (e.g., AIME 2024, AIME 2025, AMC 2023) typically favor values in the neighborhood

Table 3: Comparison of MarkovScale variants on representative reasoning benchmarks (with $N \in \{8, 16, 32, 64\}$). "Acc." denotes accuracy (%), "Toks." indicate the total number of inference tokens.

| Baseline Method | MATH-500 | | GSM8K | | AIME 2024 | | AIME 2025 | | AMC 2023 | |
|---|---|---|---|---|---|---|---|---|---|---|
| | Acc. | Toks. ($\downarrow$) | Acc. | Toks. ($\downarrow$) | Acc. | Toks. ($\downarrow$) | Acc. | Toks. ($\downarrow$) | Acc. | Toks. ($\downarrow$) |
| **Deepseek-R1-Distill-Llama-8B** | | | | | | | | | | |
| MR-Thinking ($N = 8$) | 85.4 | 21868 | **90.5** | **8552** | 43.3 | 89923 | 33.3 | 93307 | **80.0** | 34500 |
| MarkovScale$^t$ ($N = 8$) | **90.2** | 21809 | 89.7 | **8552** | 60.0 | 80956 | **40.0** | 81296 | **80.0** | **33837** |
| MarkovScale$^*$ ($N = 8$) | **90.2** | **21800** | 89.7 | **8552** | 60.0 | **80325** | **40.0** | 81296 | **80.0** | 33908 |
| MarkovScale$^0$ ($N = 8$) | **90.2** | 21868 | 89.7 | **8552** | 60.0 | 86120 | **40.0** | 80492 | **80.0** | 34500 |
| MR-Thinking ($N = 16$) | 88.2 | 43338 | 91.2 | 16916 | 50.0 | 178390 | 40.0 | 183332.0 | 82.5 | 67480.0 |
| MarkovScale$^t$ ($N = 16$) | **92.0** | 43203 | **92.0** | 16916 | 73.3 | 160233 | 43.3 | 157090 | 87.5 | 66049 |
| MarkovScale$^*$ ($N = 16$) | **92.0** | 43178 | **92.0** | **16863** | 73.3 | 158395 | 43.3 | 156790 | 87.5 | **65538** |
| MarkovScale$^0$ ($N = 16$) | **92.0** | 43338 | **92.0** | 16916 | 73.3 | 168548 | 43.3 | **155195** | 87.5 | 67480 |
| MR-Thinking ($N = 32$) | 86.4 | 86127 | 90.9 | 33819 | 60.0 | 356745 | 26.7 | 368740 | 85.0 | 133628 |
| MarkovScale$^t$ ($N = 32$) | **92.4** | 85804 | **93.2** | 33819 | 73.3 | 318040 | 43.3 | 315051 | 95.0 | 130376 |
| MarkovScale$^*$ ($N = 32$) | **92.4** | **85747** | **93.2** | **32988** | 73.3 | **314751** | 43.3 | 311657 | 95.0 | **126306** |
| MarkovScale$^0$ ($N = 32$) | **92.4** | 86127 | **93.2** | 33819 | 73.3 | 335142 | 43.3 | **306124** | 95.0 | 133628 |
| MR-Thinking ($N = 64$) | 88.2 | 171565 | 90.8 | 67478 | 56.7 | 704771 | 30.0 | 742767 | 82.5 | 266692 |
| MarkovScale$^t$ ($N = 64$) | **93.2** | 170930 | **93.9** | 67478 | 76.7 | 624233 | 43.3 | 653900 | 92.5 | 260360 |
| MarkovScale$^*$ ($N = 64$) | **93.2** | **170687** | **93.9** | **67370** | 76.7 | **617252** | 43.3 | 653900 | 92.5 | **244158** |
| MarkovScale$^0$ ($N = 64$) | **93.2** | 171565 | **93.9** | 67478 | 76.7 | 657353 | 43.3 | **632523** | 92.5 | 266692 |
| **Deepseek-R1-Distill-Qwen-7B** | | | | | | | | | | |
| MR-Thinking ($N = 8$) | 91.2 | 21396 | 88.4 | 4792.0 | 56.7 | 78432 | 40.0 | 86568 | 87.5 | 36780 |
| MarkovScale$^t$ ($N = 8$) | **94.6** | 20995 | **93.3** | 4785 | **70.0** | 35972 | **53.3** | 81231 | **92.5** | 20377 |
| MarkovScale$^*$ ($N = 8$) | **94.6** | **20817** | **93.3** | 4770 | **70.0** | **34021** | **53.3** | 80524 | **92.5** | **18617** |
| MarkovScale$^0$ ($N = 8$) | **94.6** | 21396 | **93.3** | 4789 | **70.0** | 41428 | **53.3** | 81281 | **92.5** | 36780 |
| MR-Thinking ($N = 16$) | 91.6 | 42422 | 88.3 | 9615 | 53.3 | 157507 | 43.3 | 171853 | 87.5 | 72877 |
| MarkovScale$^t$ ($N = 16$) | **95.0** | 42262 | **94.1** | 9600 | **70.0** | 64369 | **56.7** | 159080 | **95.0** | 38055 |
| MarkovScale$^*$ ($N = 16$) | **95.0** | **42108** | **94.1** | **8573** | **70.0** | **60624** | **56.7** | 156835 | **95.0** | **33421** |
| MarkovScale$^0$ ($N = 16$) | **95.0** | 42422 | **94.1** | 9609 | **70.0** | 76211 | **56.7** | 157591 | **95.0** | 72877 |
| MR-Thinking ($N = 32$) | 91.4 | 84234 | 88.4 | 19135 | 63.3 | 313463 | 33.3 | 341517 | 90.0 | 143178 |
| MarkovScale$^t$ ($N = 32$) | **95.0** | 82522 | **94.3** | 19104 | 73.3 | 254489 | 50.0 | 235833 | 95.0 | 71659 |
| MarkovScale$^*$ ($N = 32$) | **95.0** | **57993** | **94.3** | **14016** | 73.3 | 245441 | 50.0 | 235833 | 95.0 | **61585** |
| MarkovScale$^0$ ($N = 32$) | **95.0** | 84234 | **94.3** | 19122 | 73.3 | **244472** | **53.3** | 236835 | 95.0 | 143178 |
| MR-Thinking ($N = 64$) | 92.0 | 168331 | 88.2 | 38305 | 70.0 | 625850 | 40.0 | 684446 | 90.0 | 286639 |
| MarkovScale$^t$ ($N = 64$) | **95.6** | 167678 | **94.6** | 38235 | 73.3 | 501492 | 46.7 | 541722 | 95.0 | 139123 |
| MarkovScale$^*$ ($N = 64$) | **95.6** | **159118** | **94.6** | **23574** | 73.3 | 481952 | 46.7 | 541722 | 95.0 | **118222** |
| MarkovScale$^0$ ($N = 64$) | **95.6** | 168331 | **94.6** | 38277 | 73.3 | **480287** | 46.7 | **469318** | 95.0 | 286639 |
| **QwQ-32B** | | | | | | | | | | |
| MR-Thinking ($N = 8$) | 94.6 | 30378 | 95.3 | 12832 | 70.0 | 85864 | 53.3 | 95536 | **87.5** | 47288 |
| MarkovScale$^t$ ($N = 8$) | **96.6** | 13415 | **95.5** | 1358 | **73.3** | **51087** | **70.0** | **70323** | **87.5** | 13445 |
| MarkovScale$^*$ ($N = 8$) | **96.6** | **11113** | 95.4 | **1221** | **73.3** | **51087** | **70.0** | **70323** | **87.5** | 13445 |
| MarkovScale$^0$ ($N = 8$) | **96.6** | 11477 | **95.5** | 1385 | **73.3** | **51087** | **70.0** | 74022 | **87.5** | **8888** |
| MR-Thinking ($N = 16$) | 94.4 | 60801 | **95.6** | 26043 | 70.0 | 171256 | 56.7 | 189500 | 80.0 | 95434 |
| MarkovScale$^t$ ($N = 16$) | 96.6 | 24539 | **95.6** | **15016** | 76.7 | **97863** | 73.3 | 134737 | 87.5 | 22713 |
| MarkovScale$^*$ ($N = 16$) | 96.6 | **12122** | **95.6** | 23488 | 76.7 | **97863** | 73.3 | 134737 | 87.5 | 22713 |
| MarkovScale$^0$ ($N = 16$) | **96.8** | 20366 | **95.6** | 26043 | 76.7 | **97863** | 73.3 | 142663 | 87.5 | **12834** |
| MR-Thinking ($N = 32$) | 94.4 | 121565 | 95.4 | 52509 | 63.3 | 345355 | 53.3 | 383810 | 75.0 | 191141 |
| MarkovScale$^t$ ($N = 32$) | 96.4 | 46806 | **95.6** | **29705** | 80.0 | 191659 | 76.7 | 268905 | 87.5 | 40965 |
| MarkovScale$^*$ ($N = 32$) | **96.6** | **12122** | **95.6** | 45293 | 80.0 | 191659 | 76.7 | 268905 | 87.5 | 40965 |
| MarkovScale$^0$ ($N = 32$) | **96.6** | 38298 | **95.6** | 52509 | 80.0 | 191659 | 76.7 | 285287 | 87.5 | **20354** |
| MR-Thinking ($N = 64$) | 93.8 | 227838 | 94.8 | 98607 | 73.3 | 646200 | 56.7 | 719654 | 80.0 | 360068 |
| MarkovScale$^t$ ($N = 64$) | **96.6** | 85797 | **95.6** | 94012 | 83.3 | 355525 | 76.7 | 500679 | 87.5 | **73133** |
| MarkovScale$^*$ ($N = 64$) | **96.6** | **12122** | **95.6** | 78088 | 83.3 | 355525 | 76.7 | 500679 | 87.5 | **73133** |
| MarkovScale$^0$ ($N = 64$) | **96.6** | 69552 | **95.6** | 98607 | 83.3 | 355525 | 76.7 | 531857 | 90.0 | 87667 |

of 0.88–0.93, whereas easier datasets (e.g., GSM8K, MATH-500) perform best with higher settings around 0.95–0.98. For example, using the R1-Distill-Qwen-7B model with $N = 64$ and $\gamma = 10$, empirically optimal values were observed at $\tau = 0.93$ on AIME 2024, $\tau = 0.92$ on AIME 2025, $\tau = 0.93$ on AMC 2023, $\tau = 0.97$ on GSM8K, and $\tau = 0.98$ on MATH-500.

Table 5 reports scaling performance regarding the varying settings of the prior strength factor $\gamma$, using Deepseek-R1-Distill-Qwen-7B as the backbone model on the AIME 2024 benchmark. We

Table 4: Experimental results of MarkovScale* with different settings of the threshold $\tau$, using Deepseek-R1-Distill-Qwen-7B as the backbone model across different benchmarks.

| Threshold $\tau$ | MATH-500 | | GSM8K | | AIME 2024 | | AIME 2025 | | AMC 2023 | |
|---|---|---|---|---|---|---|---|---|---|---|
| | Acc. | Toks. ($\downarrow$) | Acc. | Toks. ($\downarrow$) | Acc. | Toks. ($\downarrow$) | Acc. | Toks. ($\downarrow$) | Acc. | Toks. ($\downarrow$) |
| 0.90 | 89.6 | 2,939 | 87.0 | 1,027 | 66.7 | 320,742 | 43.3 | 468,315 | 85.0 | 34,110 |
| 0.91 | 90.8 | 3,491 | 88.3 | 2,150 | 70.0 | 381,891 | 43.3 | 468,315 | 90.0 | 60,334 |
| 0.92 | 92.6 | 5,718 | 89.5 | 3,637 | 70.0 | 405,844 | **46.7** | **541,722** | 92.5 | 117,790 |
| 0.93 | 94.2 | 40,591 | 92.7 | 6,250 | **73.3** | **481,951** | 46.7 | 618,765 | **95.0** | **118,222** |
| 0.94 | 95.0 | 55,209 | 93.9 | 9,634 | 73.3 | 537,113 | 46.7 | 619,701 | 95.0 | 158,662 |
| 0.95 | 95.0 | 81,217 | 94.4 | 14,537 | 73.3 | 600,713 | 46.7 | 623,294 | 95.0 | 192,809 |
| 0.96 | 95.2 | 104,107 | 94.5 | 18,554 | 73.3 | 602,617 | 46.7 | 659,270 | 95.0 | 224,076 |
| 0.97 | 95.4 | 134,130 | **94.6** | **23,573** | 73.3 | 606,782 | 46.7 | 675,129 | 95.0 | 251,998 |
| 0.98 | **95.6** | **159,118** | 94.6 | 30,601 | 73.3 | 622,478 | 46.7 | 684,446 | 95.0 | 280,571 |
| 0.99 | 95.6 | 168,226 | 94.6 | 37,971 | 73.3 | 625,850 | 46.7 | 684,446 | 95.0 | 286,638 |

Table 5: Experimental results of different settings of the prior strength factor $\gamma$. Here, MarkovScale* employs Deepseek-R1-Distill-Qwen-7B on the AIME 2024 benchmark.

| Threshold $\tau$ | $\gamma = 1$ | | $\gamma = 4$ | | $\gamma = 7$ | | $\gamma = 10$ | | $\gamma = 20$ | |
|---|---|---|---|---|---|---|---|---|---|---|
| | Acc. | Toks. ($\downarrow$) | Acc. | Toks. ($\downarrow$) | Acc. | Toks. ($\downarrow$) | Acc. | Toks. ($\downarrow$) | Acc. | Toks. ($\downarrow$) |
| 0.80 | 60.00 | 47,360 | 53.33 | 9,884 | 53.33 | 9,884 | 53.33 | 9,884 | 53.33 | 9,884 |
| 0.81 | 60.00 | 47,360 | 53.33 | 9,884 | 53.33 | 9,884 | 53.33 | 9,884 | 53.33 | 9,884 |
| 0.82 | 60.00 | 47,360 | 53.33 | 9,884 | 53.33 | 9,884 | 53.33 | 9,884 | 53.33 | 9,884 |
| 0.83 | 63.33 | 68,998 | 53.33 | 9,884 | 53.33 | 9,884 | 53.33 | 9,884 | 53.33 | 9,884 |
| 0.84 | 63.33 | 69,498 | 53.33 | 9,884 | 53.33 | 9,884 | 53.33 | 9,884 | 53.33 | 9,884 |
| 0.85 | 63.33 | 70,511 | 53.33 | 9,884 | 53.33 | 9,884 | 53.33 | 9,884 | 53.33 | 9,884 |
| 0.86 | 63.33 | 97,876 | 53.33 | 9,884 | 53.33 | 9,884 | 53.33 | 9,884 | 60.00 | 91,438 |
| 0.87 | 63.33 | 98,403 | 53.33 | 9,884 | 53.33 | 9,884 | 53.33 | 9,884 | 66.67 | 256,804 |
| 0.88 | 63.33 | 98,403 | 53.33 | 9,884 | 53.33 | 9,884 | 53.33 | 37,725 | 73.33 | 423,459 |
| 0.89 | 63.33 | 98,547 | 53.33 | 9,884 | 53.33 | 9,884 | 66.67 | 224,497 | 73.33 | 483,868 |
| 0.90 | 63.33 | 99,068 | 53.33 | 9,884 | 53.33 | 37,725 | 66.67 | 320,742 | 73.33 | 563,873 |
| 0.91 | 63.33 | 131,530 | 53.33 | 9,884 | 66.67 | 200,576 | 70.00 | 381,891 | 73.33 | 583,646 |
| 0.92 | 63.33 | 131,964 | 53.33 | 9,884 | 66.67 | 257,174 | 70.00 | 405,844 | 73.33 | 595,669 |
| 0.93 | 63.33 | 162,589 | 53.33 | 9,884 | 70.00 | 353,703 | 73.33 | 481,951 | 73.33 | 604,932 |
| 0.94 | 63.33 | 162,589 | 53.33 | 9,884 | 70.00 | 353,703 | 73.33 | 537,113 | 73.33 | 607,838 |
| 0.95 | 63.33 | 162,589 | 53.33 | 9,884 | 70.00 | 381,891 | 73.33 | 600,713 | 73.33 | 615,435 |
| 0.96 | 63.33 | 225,103 | 53.33 | 69,710 | 73.33 | 463,866 | 73.33 | 602,617 | 73.33 | 622,944 |
| 0.97 | 63.33 | 225,103 | 60.00 | 117,653 | 73.33 | 501,492 | 73.33 | 606,782 | 73.33 | 625,850 |
| 0.98 | 66.67 | 250,354 | 63.33 | 185,195 | 73.33 | 547,403 | 73.33 | 622,478 | 73.33 | 625,850 |
| 0.99 | 66.67 | 250,354 | 66.67 | 269,499 | 73.33 | 618,059 | 73.33 | 625,850 | 73.33 | 625,850 |

adopt $\gamma = 10$, a representative mid-range setting. Empirical results in the table support this choice: low prior strengths (e.g., $\gamma = 1$ or $\gamma = 4$) lead to markedly poorer performance, with peak accuracies of 63.33% for $\gamma = 1$ and 60.00% for $\gamma = 4$, compared to 73.33% for $\gamma = 10$. The sensitivity analysis also shows that tuning $\gamma$ upward from very small values generally yields consistent accuracy gains, while overly weak priors incur substantial accuracy degradation across nearly all thresholds. Together, these trends validate $\gamma = 10$ as a robust and well-calibrated setting.

