# OpenReview forum: "MarkovScale: Towards Optimal Sequential Scaling at Inference Time"
_ICLR.cc/2026/Conference — Submitted to ICLR 2026_

### Official Review · Reviewer_NLpN · 2025-10-20

**Soundness:** 3
**Presentation:** 2
**Contribution:** 3
**Rating:** 6
**Confidence:** 3

**Summary:**

The authors pose sequential test time scaling for LLMs as a two state Markov chain: Given the present completion from the model (or the question initially) either the model generates a correct (C) or an incorrect (W) completion in the next step of inference. By analyzing this Markov chain they identify the conditions under which scaling is beneficial, neutral and detrimental. They also relate the expected convergence to the probability of the model moving from C->W or from W->C which then enables them to determine upper and lower bounds on performance when scaling in this way. Using this they develop a series of methods that aim to determine when to use scaling, and how much scaling to use. They then compare their methods with other test-time scaling methods, showing it has favorable performance on 3 models and 5 test sets.

**Strengths:**

- This work is timely and addresses a key question about test-time scaling, namely how much to do and in what circumstances it is beneficial to do so
- The performance bounds are nice and give a benchmark against which to compare methods
- Despite some problems with the exposition the central idea in this paper is quite elegant and uncomplicated.

**Weaknesses:**

- Figure 3 could do with some improvement. I think a bar chart or some other chart that doesn't imply an interpolation between benchmarks might be more appropriate.
- The exposition in section 3.3 is a bit sloppy. For instance, where does this theoretical bias term originate from? What does $q$ represent (a question I'm guessing)? Is $p$ in (6), (7) and (8) $p_0$ or $p_i$?
- I think the framing around section 3.3 could do with some justification: I'm a bit skeptical about model capability and problem difficulty being disentangled in this way. Surely the models capability is also related to the zero shot probability of correctness, just as the problems difficulty is related to $a$ and $b$? Additionally you frame $a$ and $b$ as being intrinsic to the model, yet you clearly estimate it separately for each dataset (or maybe example if $q$ represents a single question) or how else would the results in figure 2 be different? I would have like a bit more clarity about exactly what $a$ and $b$ represent and how they were estimated.
- I would have liked to have seen baseline results with no test time scaling.

**Questions:**

All of your methods appear to perform quite similarly, especially on the smaller models, are there any reasons to prefer one over the others?

---

> ### Author Response · Authors · 2025-11-27
> **Responses to Reviewer NLpN**
>
> We sincerely appreciate your positive evaluation of our work, particularly concerning the key problem addressed, the novel idea proposed, and the performance bound derived. Thank you for your constructive comments and suggestions. Below, we provide additional clarifications and experimental results to address your remaining concerns.
>
> W1: **Improvement of Figure 3.**
>
> Thank you for your suggestion. In the revised manuscript, Figure 3 has been revised and is now presented as a range chart. This visualization is better suited for comparing the performance of our distinct scaling groups (e.g., Beneficial Group, MarkovScale, etc.) and avoids the misleading implication of continuity.
>
> W2: **The exposition in section 3.3 is a bit sloppy.**
>
> Thank you for pointing out the ambiguities. We clarify that: 1) The bias term originates from the inherent simplification in modeling a complex, sequential reasoning process using the Markov chain. Since the Markov assumption ignores potential long-range dependencies or non-stationary effects in the sequential steps, the bias term aggregates the unmodeled dependencies, prediction error margins, and random fluctuations that are not captured by the state transition probabilities. This term is necessary to align the theoretical model with the observed empirical performance. 2) We confirm that $q$ represents a question. 3) The variable $p$ in Equations (6)-(8) should indeed be the initial probability $p_0$. We have updated those equations for clear notations.
>
> We have improved the presentation of relevant notations throughout **Section 3.3** in the revised manuscript.
>
>
>
> W3: **The framing around section 3.3 could do with some justification.**
>
> Thank you for your valuable feedback.
>
> (1) We agree that model capability and problem difficulty are related. Our approach models their joint dependency rather than fully disentangling them. The parameters $a$ (Correct $\to$ Incorrect) and $b$ (Incorrect $\to$ Correct) represent the model's reliability and potential for self-correction during sequential refinement steps given specific questions. The zero-shot probability $p_0$ reflects the specific difficulty of a given question $q$ for that given model.
>
> (2) Estimation process of $a$ and $b$.
>
> For MarkovScale$^{*}$, we train separate MLP layers as estimators for predicting $a$ and $b$. To construct the training dataset, we run naive sequential scaling (multi-round thinking) on the training set and statistically count state transitions as follows:
>
> $a$ (Correct $\to$ Incorrect): The ratio of transitions from a correct state to an incorrect answer.
>
> $b$ (Incorrect $\to$ Correct): The ratio of transitions from an incorrect state to a correct answer.
>
> This would create question-probability pairs for training the estimators, which are then used to predict per-question transition probabilities. While the estimators predict question-specific probabilities, we observed high variance in per-question estimates. Therefore, we average the predicted $a$ and $b$ across the dataset to obtain stable, dataset-level priors during inference.
>
> For MarkovScale$^{0}$, there are no estimators for $a$ and $b$. We apply a small number of inference rounds as a “warm-up” phase for questions within each dataset, where we get a considerably large pool of state transitions over the dataset. Then, we statistically count the ratio of transitions from a correct state to an incorrect answer ($a$) and the ratio of transitions from an incorrect state to a correct answer $b$. This serves as an approximated empirical estimation of the transition probabilities, which will be used in MarkovScale$^{0}$ during inference.
>
> To provide greater clarity, we have justified the detailed computation processes in Section 3.3 of the revised manuscript.
>
>
>
> W4: **Baseline results with no test time scaling.**
>
> We have reported baseline results without test-time scaling in **Table 1** of the revised manuscript. These results show that base models without test-time scaling generally perform significantly worse than those with test-time scaling.
>
>
>
> W5: **Reasons to prefer which variants for the proposed method.**
>
> Thank you for raising this insightful question. We have added a full comparison between different variants of MarkovScale in **Appendix C**. We observe that while MarkovScale$^{0}$ achieves strong accuracy, it consistently incurs higher inference-time token costs. In comparison, MarkovScale* achieves comparable high accuracy while dramatically reducing token consumption, demonstrating that this training variant successfully guides the optimal stopping policy. This allows for **demand-based choice**: If you want a completely training-free approach, choose MarkovScale$^{0}$. If you don’t mind spending some compute upfront to train an estimator and require better inference efficiency, choose MarkovScale*. Both options deliver excellent accuracy.

---

### Official Review · Reviewer_puxx · 2025-10-30

**Soundness:** 3
**Presentation:** 3
**Contribution:** 3
**Rating:** 6
**Confidence:** 2

**Summary:**

This paper proposes a new algorithm, based on a discrete Markov process, for inference time scaling of LLMs via sequental scaling. I am no expert in LLMs or methods research on top of LLMs, so I don't have high confidence in my review. What is more, I am short on time due to the semester start. Apologies if my reviews are a bit short. I am happy to engage in reviewer discussion should be concerns not be clear.
That said, I think the suggested approach appears sensible, easy to understand and verify, and leads to improved results compared to baselines. So, from my far away perspective, I think this paper is a relevant contribution to the field.

**Strengths:**

- Clear theoretical framework.
- Easy to understand approach yet providing good empirical accuracy.
- Clear and consistent improvements against many benchmarks.

**Weaknesses:**

- I am too far away from the field to judge this in detail

**Questions:**

- The notation around EQ2 looks a bit messy. Sometimes there is an index i, sometimes there is not. a P seems to be missing in EQ2. Can you double check and fix the notation to be consistent?

---

> ### Author Response · Authors · 2025-11-27
> **Response to Reviewer puxx**
>
> We sincerely thank you for your positive feedback on our work. We address your concern below to improve the clarity of this paper.
>
> **Q1: The notation around EQ2 looks a bit messy. Sometimes there is an index i, sometimes there is not. a P seems to be missing in EQ2.**
>
> Thank you for your careful reading and for raising this notation concern. In Eq. (2), we have revised the expression from $\big[X_i=C, X_i=W\big]$ to $\big[\mathbb{P}(X_i=C), \mathbb{P}(X_i=W)\big]$ to make it explicit that these terms represent event probabilities. Furthermore, the index i is intended as a superscript exponent, not a subscript reference, and denotes repeated powers rather than node indexing. To eliminate any remaining ambiguity, we now include an intermediate expansion that unrolls the probability product, clarifying that the transition matrix P is multiplied i times. The notation has been updated in the revised manuscript **(EQ2 in Section 3.1)** for improved clarity and readability.

---

### Official Review · Reviewer_SX1J · 2025-10-31

**Soundness:** 2
**Presentation:** 3
**Contribution:** 3
**Rating:** 6
**Confidence:** 3

**Summary:**

The paper first makes an observation that parallel scaling tends provide better performance but is token efficient compared to sequential scaling approaches. The paper aims to optimize the sequential scaling.
Given the inference-time scaling, it seems that the paper is trying to formulate the problem of scaling as a two-state Markov process. The work uses this formulation for the correctness probability to get the performance bounds: neutral, upper, and lower and use that as the theoretical basis to directly estimating the convergence accuracy.
MarkovScale proposed in the paper has several variants: (1) gating strategy, (2) MAP-based optimal scaling, and (3) training-free version.
The paper evaluates the MarkovScale approach on combinations of several models and benchmark to compare against several methods including ones based on budget, early stopping, self-consistency, etc.
The paper demonstrates that it achieves better results (accuracy) given same token consumption.

**Strengths:**

* The paper is tacking a very important topic of token-efficient inference-time scaling.
* It seems the paper is trying to move from previous heuristic based approaches to an approach that is a little more backed by theoretical formulation which seems nice.

**Weaknesses:**

* Formulation seems rather oversimplified as overall reasoning process in inference-time scaling is not easy to boil down to simply correct vs incorrect. It is literally a reasoning process where things can go south then use that as a context to later converge on a better outcome. However, the oversimplification of the formulation seems to understate the significance of this.
* Seems rather unclear how the transition probabilities and zero-shot probability are computed in the MarkovScale.

**Questions:**

* Can you please elaborate how transition probabilities and zero-shot probability are computed in the MarkovScale. Some details would be great.
* Is there any projection to how this can be generalized to different models of different scale? It seems that the models evaluated in the paper are pretty small.
* Figure 1 stops the evaluation at around 700k tokens. What happens after that would also be interesting data to show whether the work (as well as the other approaches) are really controlling the inference-time scaling in an optimal manner.
* It is also interesting to see in Table 1 that MarkovScale0 seems to perform better than other variants a few times. Is this just error margin? Given that the MarkovScale based on MAP could be assumed to provide a good performance, does this mean that MLP was not trained enough? It seems that it is difficult to fully rule out the overfitting?
* Is there a good results showing how hyperparameters were determined?

---

> ### Author Response · Authors · 2025-11-27
> **Responses to Reviewer SX1J (Part 1)**
>
> We sincerely thank you for your thoughtful and constructive feedback on our work, and we greatly appreciate your positive evaluation of our theoretical formulation. Below, we address your concerns with additional clarifications and experimental results.
>
> **W1: Formulation of inference-time scaling seems rather oversimplified.**
>
> Thank you for your insightful observation. We fully agree that inference-time scaling is not strictly binary, as its reasoning process is a dynamic trajectory where intermediate errors can indeed serve as context for subsequent self-correction (a non-monotonic process). However, we hope to highlight that:
>
> (1) **Our 2-state Markovian formulation serves as a necessary first-order approximation to make the theoretical analysis of sequential scaling tractable.** Despite its simplicity, this abstraction proves to be highly effective and successfully captures the macroscopic dynamics of scaling, such as the phenomenon of performance decay, and accurately characterizes the theoretical bounds (see Sections 3.3 & 3.4). Crucially, **the validity of this formulation is empirically supported** by the fact that the derived adaptive scaling strategy MarkovScale achieves SOTA performance.
>
> (2) While the current model aggregates the unmodeled complexities (e.g., the fluctuations mentioned) into the bias term $\sigma$, we agree that a more granular model is a promising direction. In future work, we plan to generalize this framework to a **Multi-state Markov Decision Process**, where intermediate states are characterized by verifier confidence scores. This would allow for a precise modeling of the internal rectification process distinct from the final outcome.
>
> To improve clarity, we have improved the formulation with necessary clarifications in **Section 3** of the revised manuscript.
>
>
>
> **W2: Unclear how the transition probabilities and zero-shot probability are computed in the MarkovScale.**
>
> We clarify that these probabilities are computed as follows:
>
> **(1) Computation of Zero-shot Probability ($p_0$)**
>
> For MarkovScale$^{*}$ (learning-based), we employ a two-stage approach combining offline learning with online refinement.
>
> *Offline Learning of Priors*: We train an MLP on top of frozen LLM embeddings using a constructed dataset of question-probability pairs $(q, \bar{p})$. Here, $\bar{p}$ is the aggregated correctness rate of sampled answers for question $q$. The MLP predicts an initial prior $\hat{p}_0(q)$.
>
> *Online Refinement of Posteriors*: During inference, $\hat{p}_0$ serves as the prior for a Beta distribution. As the model generates new answers, we perform Maximum-A-Posteriori (MAP) estimation to update $p_0$ using scores from a verifier.
>
> For MarkovScale$^{0}$, which is training-free without offline predictors, we initialize $p_0$ with a heuristic fixed value (e.g., 0.9 for standard datasets like GSM8K/MATH-500; 0.8 for harder tasks like AIME). Similarly, we then employ MAP to continuously refine $p_0$ during inference.
>
> **(2) Computation of Transition Probabilities ($a$ and $b$)**
>
> For MarkovScale*, we train separate MLP layers as estimators for predicting $a$ and $b$. To construct the training dataset, we run naive sequential scaling (multi-round thinking) on the training set and statistically count state transitions as follows:
>
> $a$ (Correct $\to$ Incorrect): The ratio of transitions from a correct state to an incorrect answer.
>
> $b$ (Incorrect $\to$ Correct): The ratio of transitions from an incorrect state to a correct answer.
>
> This would create question-transition probability pairs for training the estimators, which are then used to predict per-question transition probabilities. While the estimators predict question-specific probabilities, we observed high variance in per-question estimates. Therefore, we average the predicted $a$ and $b$ across the dataset to obtain stable, dataset-level priors during inference.
>
> For MarkovScale$^{0}$, there are no estimators for $a$ and $b$. We apply a small number of inference rounds as a “warm-up” phase for questions within each dataset, where we get a considerably large pool of state transitions over the dataset. Then, we statistically count the ratio of transitions from a correct state to an incorrect answer ($a$) and the ratio of transitions from an incorrect state to a correct answer $b$. This serves as an approximated empirical estimation of the transition probabilities, which will be used in MarkovScale$^{0}$ during inference.
>
> We have improved by adding more details about the computation for the zero-shot probability ($p_0$) and transition probabilities ($a, b$) in **Section 4.2** of the revised manuscript.

---

> ### Author Response · Authors · 2025-11-27
> **Responses to Reviewer SX1J (Part 2)**
>
> **W3: How this method is generalized to different models of different scales.**
>
> Thank you for raising this important question of scalability. We clarify that we have included results for larger-scale models (e.g., QwQ-32B) in Table 1. Our results show that MarkovScale demonstrates robust performance across R1-Distill-Qwen-2.5-7B, R1-Distill-Llama-3.1-8B, and QwQ-32B. This consistency across different model families and scales (from 7B to 32B) validates that the proposed method generalizes effectively beyond small models. We have improved the discussion of this aspect in **Section 5.2** of the revised manuscript.
>
>
>
> **W4: What happens when continual scaling after 700k tokens in Figure 1?**
>
> We appreciate your suggestion to investigate the long-horizon behavior of scaling. To this end, we extended the sequential scaling evaluation to 1,400k tokens (doubling the original 700k range) on the AIME 2024 dataset using two backbones: Deepseek-R1-Distill-Qwen-7B and Deepseek-R1-Distill-Llama-8B. The results validate the following findings: **1) Sustained Superiority:** MarkovScale continues to significantly outperform the sequential scaling baseline (MR-Thinking) at **1,400k tokens**. On Qwen-7B, MarkovScale achieves 70.0% accuracy vs. 53.3% for the baseline (a +16.7% margin); On Llama-8B, MarkovScale achieves 66.7% accuracy vs. 60.0% for the baseline. **2) Efficiency & Optimality:** Regarding the “optimal manner” of control, MarkovScale achieved the 70% peak on Qwen-7B while reducing total token consumption by 67.3% compared to the baseline’s saturation point. This indicates that MarkovScale effectively controls the scaling process at large horizons. We will include more extended results in the final version.
>
>
>
> **W5: Comparison between MarkovScale$^{0}$ and MarkovScale$^{*}$ with MAP.**
>
> Thank you for your insightful observation. The fact that the training-free MarkovScale$^{0}$ (which uses a simple uniform prior for $p_0$) sometimes outperforms other variants validates **the robustness and generality of the core Markovian formulation itself**. The primary objective of the trainable variant (MarkovScale*) is not merely to achieve peak accuracy, but to control the scaling process in an optimal and efficient manner. As reported in **Appendix C**, while MarkovScale$^{0}$ achieves strong accuracy, it usually incurs higher inference-time token costs than MarkovScale*. In comparison, **MarkovScale$^{*}$ achieves comparable high accuracy while greatly reducing token consumption**, demonstrating that this trainable variant successfully guides the optimal stopping policy, which is the key contribution. This also leaves room for **demand-based choice**: If you want a completely training-free approach, choose MarkovScale$^{0}$. If you don’t mind spending some compute upfront to train an estimator and require better inference efficiency, choose MarkovScale*. Both options deliver excellent accuracy.
>
> We have incorporated the comparative results for accuracy and token consumption of the different MarkovScale variants into **Appendix C** of the revised manuscript.
>
>
>
> **W6: Results showing how hyperparameters were determined.**
>
> Thank you for raising this thoughtful question. Our experiments primarily involve two key hyperparameters:
>
> **(1) The Stopping Threshold ($\tau$).** This parameter dictates the required confidence probability for stopping the inference process. The sensitivity study detailed in **Table 4**, reveals that the optimal choice for $\tau$ is correlated with benchmark difficulty. Specifically, harder tasks (e.g., AIME 2024, AIME 2025, AMC 2023) generally favor values between $0.88$ and $0.93$, while easier datasets (e.g., GSM8K, MATH-500) perform best with higher settings around $0.95$ to $0.98$.
>
> **(2) The Prior Strength Factor ($\gamma$).** Used exclusively in MarkovScale$^{*}$, $\gamma$ controls the weight assigned to the offline prior ($\hat{p}_0$) during the online MAP update of the Beta distribution parameters. We adopt $\gamma=10$ as a robust, empirically determined mid-range setting. The results in **Table 5** strongly support this choice: low prior strengths (e.g., $\gamma=1$ or $\gamma=4$) result in markedly poor performance (peak accuracies of $63.33\%$ for $\gamma=1$ and $60.00\%$ for $\gamma=4$), compared to the $73.33\%$ accuracy achieved with $\gamma=10$. The sensitivity analysis further demonstrates that increasing $\gamma$ from very small values consistently yields accuracy gains, confirming $\gamma=10$ as a well-calibrated and robust setting.
>
> We have incorporated the detailed results of these parameter analyses (Tables 4 and 5) into **Appendix D** of the revised manuscript.

---

### Meta-Review · Area_Chair_VKzR · 2026-01-10

**Summary:**

This paper proposes MarkovScale, a framework that models sequential test-time scaling as a two-state Markov process and derives closed-form conditions and bounds meant to guide when and how long to scale. Building on this formulation, it introduces practical variants (training-free and learned/MAP-based) and reports improved accuracy–token trade-offs across several LLM backbones and reasoning benchmarks.

The idea is timely and the paper has clear strengths: the Markov abstraction is simple and communicable, the bounds provide a useful lens for thinking about when sequential scaling helps versus hurts, and the experiments suggest consistent gains at matched token budgets. However, I do not recommend acceptance because the central “optimality” narrative currently outpaces what is justified by the modeling and evidence. The two-state formulation appears too coarse for real multi-step reasoning dynamics, and the theory relies on quantities (zero-shot probability and transition rates) that are not cleanly identified in practice. In the discussion, the authors clarify that per-question transition estimates are high variance and therefore they fall back to dataset-level priors or warm-up counting, and that the learned components and thresholds are tuned empirically. This makes the approach feel closer to a principled heuristic controller than an optimal policy with robust guarantees. As a result, it is difficult to assess generality and robustness across datasets, verifiers, prompting formats, and model families, and it is not clear that the claimed optimality bounds tightly predict real long-horizon behavior rather than providing post-hoc explanation. With all reviews landing around borderline territory, the paper is borderline, but I do not think it is good enough at this point.

**Reviewer Concerns:**

Addressed by the rebuttal and revision:
The rebuttal meaningfully improves clarity on implementation details that were previously underspecified. In particular, it explains how the method estimates the initial correctness probability and transition probabilities (via offline MLP priors and online MAP refinement, or training-free warm-up counting), and it responds to requests for longer-horizon scaling behavior by extending the token budget beyond the original figure. The authors also fixed notation issues around Eq. (2) and acknowledged and corrected several presentation ambiguities (what variables represent, inconsistent symbols). The response also clarifies that results include larger backbones (up to a 32B model) and provides additional sensitivity analysis for key hyperparameters.

Still outstanding:
The primary remaining concern is foundational rather than missing details: the two-state Markov abstraction is likely an oversimplification of iterative reasoning and self-correction, and the paper does not convincingly validate that the Markov assumptions (stationarity, two-state sufficiency) hold in a way that supports the strength of the “optimal” framing. The introduction of an explicit bias term to account for unmodeled effects underscores this gap, but the paper does not characterize when this bias is small or how it impacts the bounds. Second, the estimation strategy for transition rates appears dataset-dependent in practice (averaging predicted values or warm-up counting), which raises questions about transfer and robustness. Third, the empirical section would benefit from clearer disentanglement of what drives the gains (the Markov formulation itself versus learned priors, verifier choice, and threshold tuning), and from stronger “no scaling” baselines and stress tests across different verifiers and prompt styles. Overall, the rebuttal helps with clarity and some completeness, but it does not fully resolve the mismatch between the model’s simplicity and the paper’s broad optimality claims.

**Reviewer Scores:**

Reviewer SX1J (rating 6, confidence 3)
Likely stays at 6. The authors directly answered the reviewer’s main questions about computing probabilities, larger-model results, extended horizon behavior, and hyperparameter selection. That should increase confidence in the implementation and reduce ambiguity. However, SX1J’s substantive worry that the formulation is oversimplified is only partially addressed (acknowledged as a “first-order approximation”), so I do not expect a decisive upward shift.

Reviewer puxx (rating 6, confidence 2)
Likely stays at 6. This reviewer was broadly positive but explicitly low-confidence and “far from the field,” with only minor notation concerns. Since the notation issue was fixed, they might feel slightly more comfortable, but given their stated limited expertise, I would not expect a material score change.

Reviewer NLpN (rating 6, confidence 3)
Likely stays at 6 or drops below to 5. The rebuttal addresses many presentation issues (figure style, notation, what variables represent) and adds the missing “no scaling” baseline, which should help. But NLpN’s skepticism about the framing in Section 3.3 and about what exactly is intrinsic to the model versus dataset-specific estimation remains only partially resolved. They might be more satisfied with clarity, but the conceptual concern likely keeps this in weak-accept/weak-reject territory.

---

### Decision · Program_Chairs · 2026-01-26

Reject